# Influence of *Benincasa hispida* Peel Extracts on Antioxidant and Anti-Aging Activities, including Molecular Docking Simulation

**DOI:** 10.3390/foods12193555

**Published:** 2023-09-25

**Authors:** Pimpak Phumat, Siripat Chaichit, Siriporn Potprommanee, Weeraya Preedalikit, Mathukorn Sainakham, Worrapan Poomanee, Wantida Chaiyana, Kanokwan Kiattisin

**Affiliations:** 1Faculty of Pharmacy, Chiang Mai University, Chiang Mai 50200, Thailand; pimpak.p@cmu.ac.th (P.P.); siripat.chaichit@cmu.ac.th (S.C.); siriporn_pot@cmu.ac.th (S.P.); 2Department of Cosmetic Sciences, School of Pharmaceutical Sciences, University of Phayao, Phayao 56000, Thailand; weeraya.pr@up.ac.th; 3Department of Pharmaceutical Sciences, Faculty of Pharmacy, Chiang Mai University, Chiang Mai 50200, Thailand; mathukorn.s@cmu.ac.th (M.S.); worrapan.p@cmu.ac.th (W.P.); wantida.chaiyana@cmu.ac.th (W.C.)

**Keywords:** *Benincasa hispida*, peel extract, rutin, antioxidant, anti-aging, anticollagenase, antihyaluronidase, HET-CAM, docking simulation

## Abstract

*Benincasa hispida* peel, a type of postconsumer waste, is considered a source of beneficial phytochemicals. Therefore, it was subjected to investigation for biological activities in this study. *B. hispida* peel was extracted using 95% *v*/*v*, 50% *v*/*v* ethanol and water. The obtained extracts were B95, B50 and BW. B95 had a high flavonoid content (212.88 ± 4.73 mg QE/g extract) and phenolic content (131.52 ± 0.38 mg GAE/g extract) and possessed high antioxidant activities as confirmed by DPPH, ABTS and lipid peroxidation inhibition assays. Moreover, B95 showed inhibitory effects against collagenase and hyaluronidase with values of 41.68 ± 0.92% and 29.17 ± 0.66%, which related to anti-aging activities. Via the HPLC analysis, one of the potential compounds found in B95 was rutin. Molecular docking has provided an understanding of the molecular mechanisms underlying the interaction of extracts with collagenase and hyaluronidase. All extracts were not toxic to fibroblast cells and did not irritate the hen’s egg chorioallantoic membrane, which indicated its safe use. In conclusion, *B*. *hispida* peel extracts are promising potential candidates for further use as antioxidant and anti-aging agents in the food and cosmetic industries.

## 1. Introduction

Ultraviolet (UV) radiation is a potent initiator of reactive oxygen species (ROS) generation in the skin, leading to oxidative imbalance [1]. ROS can impair the skin barrier functions by altering squalene, cholesterol and unsaturated lipids, which are essential components of the skin’s structure. The effect of skin oxidative stress caused by excessive ROS leads to the initiation of several skin problems such as atypical pigmentation, skin inflammation, increasing sebum secretion and increased levels of oxidized lipids, including skin aging [2,3]. Collagen, elastin and hyaluronic acid (HA) are biological compounds that help promote healthy and youthful skin. Collagen and elastin, the proteins abundant in the dermal layer, play essential roles in maintaining skin flexibility, elasticity and integrity, whereas HA, a glucose-based polymer, in the dermis and the epidermis layers mainly promotes skin rejuvenation and moisture [4,5]. The over-accumulation of ROS leads to the activation of dermal enzymes including collagenase, elastase and hyaluronidase, which in turn degrade collagen, elastin and HA [6]. Thus, various antioxidants simplify skin protection against oxidative damage caused by ROS. Both chemical and natural antioxidants are investigated and incorporated into numerous skin care products to mitigate skin damage. Medicinal plants have been reported to provide potential antioxidation activity through various mechanisms. For example, *Thunbergia laurifolia* Lindl. leaf extracts have demonstrated antioxidation activities through radical scavenging and inhibiting peroxidation mechanisms [7], and *Acacia concinna* Linn. pod extracts exhibited scavenging activity against free radicals [8].

*Benincasa hispida* (Thunb.) Cogn, a winter melon also known as Fuk-Kiew in Thailand, is an edible plant in the family Cucurbitaceae, generally found in Asia, especially in northern Thailand. Scientific reports suggest that *B. hispida* is rich in nutrients, minerals, vitamins and phytochemical compounds [9]. The bioactive compounds in *B*. *hispida* extract include phenolics, triterpenoids, flavonoids, glycosides, carotenes and β-sitosterin [10,11]. Related studies have reported that *B. hispida* exhibits pharmacologic effects including antioxidation, anti-angiotensin-converting enzyme, anti-inflammation and antibacterial effects. [12,13]. In addition, the efficacy and phytochemical components of the pulp, fruit and seeds of *B*. *hispida* were determined [11]. Gallic acid, catechin and ascorbic acid, which are phenolic compounds found in the fruit of *B*. *hispida*, exhibit antioxidant activity by reducing free radicals [14,15,16].

The fruit of *B. hispida* is broadly cylindrical in shape, as shown in Figure 1A. Generally, the edible pulp of the fruit is used in cooking and the production of various food products such as tea and juice. Thus, the peel of the fruit then becomes a form of postconsumer waste, as shown in Figure 1B. Interestingly, *B. hispida* juice has gained popularity as a consumer product. This trend has resulted in an increased production of *B. hispida* peel, which is considered a waste product in the food industry. Notably, *B. hispida* fruit can be stored for several months, possibly because of the presence of a protective wax coating on the peel. Thus, investigating the constituents within the peel responsible for the ability to protect the pulp from external environmental influences as shown in Figure 1B would be intriguing. Therefore, the present study focuses on investigating the antioxidant and anti-aging properties of *B. hispida* peel, conducting simulations of the possible docking mechanisms of the potential compounds in the peel and performing safety tests. These research methods are designed to transform *B. hispida* peel, traditionally considered a waste product, into a high-value material for potential applications as an innovative cosmetic ingredient or another innovative material using its biological activities and the mechanisms of action determined through docking simulations.

## 2. Materials and Methods

### 2.1. Materials

The plant material used in this study was fresh *B. hispida* fruit, collected from Phrae Province, Thailand. The plant was identified, and its voucher specimen was stored in the Herbarium of the Faculty of Pharmacy, Chiang Mai University, Thailand.

The reagents and chemicals were of analytical grade. 2,2-Diphenyl-1-picrylhydrazyl (DPPH) and 2,2-azino-bis(3-ethylbenzthiazoline)-6-sulfonic acid (ABTS) were purchased from Fluka (Buchs, Switzerland). Folin–Ciocalteu reagent was purchased from Merck (Darmstadt, Germany). 6-Hydroxy-2,5,7,8-tetramethyl chroman-2-carboxylic acid (Trolox), gallic acid, ascorbic acid, kojic acid, epigallocatechin-3-gallate (EGCG), hyaluronic acid, linoleic acid, rutin, 3-(4,5-dimethylthiazolyl-2)-2,5-diphenyl tetrazolium bromide (MTT) and 2,4,6-Tris(2-pyridyl)-s-triazine (TPTZ) were purchased from Sigma (St. Louis, MO, USA). Dulbecco’s Modified Eagle medium (DMEM) and penicillin–streptomycin were acquired from Invitrogen (Grand Island, NY, USA). Fetal bovine serum (FBS) and bovine serum albumin (BSA) were obtained from Biochrom AG (Berlin, Germany). Ethanol and dimethyl sulfoxide (DMSO) were purchased from Labscan Asia Co., Ltd. (Bangkok, Thailand). HPLC-grade acetonitrile was purchased from Merck (Darmstadt, Germany). Lipopolysaccharide (LPS) was purchased from Sigma-Aldrich (Darmstadt, Germany), and sodium carbonate, aluminum chloride and sodium nitrite were purchased from United Chemical & Trading Co., Ltd. (Chiang Mai, Thailand). Other chemicals and solvents were of the highest grade available.

### 2.2. Plant Extraction

The procedure was performed following a related study with modification [17]. Briefly, the fresh peel of *B. hispida* fruit was dried in a hot air oven (Memmert, WI, USA) at 50 °C for 24 to 48 h and then ground to a fine powder. The obtained plant powder was extracted with various solvents using ultrasound-assisted extraction (UAE). For extraction, 95% *v*/*v* ethanol, 50% *v*/*v* ethanol and deionized (DI) water were used as solvents. In each solvent extraction, an ultrasonic bath (Elma Schmidbauer GmbH, Singen, Germany) was used to generate the ultrasound effect for 30 min at 65.0 ± 1.0 °C. Each UAE was performed in triplicate. The extracted mixture (plant powder mixed with solvent) was filtered through a Whatman No. 1 filter paper. The filtrates obtained from 95% *v*/*v* ethanol and 50% *v*/*v* ethanol were subjected to a rotary vacuum evaporator (M.A-3S, Eyela, Tokyo, Japan) to remove the solvent, whereas the filtrate obtained from DI water was subjected to a lyophilizer (Christ Beta 2-8 LD plus, Osterode am Harz, Germany). After the solvents were completely removed, the extracts from 95% *v*/*v* ethanol (B95), 50% *v*/*v* ethanol (B50) and DI water (BW) were obtained, and all extracts were stored at 4 °C until use. The %yield of each extract was calculated using the following equation:%yield = (E/P) × 100(1)
where E represents the weight of the extract and P represents the weight of the plant powder used for extraction.

### 2.3. Determination of Total Flavonoid Content

Total flavonoid content (TFC) was determined using aluminum chloride colorimetry following our related study [18]. All *B. hispida* extracts were prepared at a concentration of 1.0 mg/mL in DI water. The extract with 100 µL was mixed with 30 µL of 5% *v*/*v* sodium nitride. After 5 min, 50 µL of 2% *v*/*v* aluminum chloride was added. The solution mixture was left for 6 min, followed by adding 50 µL of 1N sodium hydroxide. The mixture was stored in the dark for 10 min at room temperature before analysis. The mixed solution was measured for absorbance at 510 nm using a UV spectrophotometer (UV-2600i, Shimadzu, Kyoto, Japan) against a blank without the extract. The standard curve for TFC determination was constructed using a quercetin (QE) solution (0.0 to 1.0 mg/mL) under the same procedure as the tested *B. hispida* extract. The TFC value was expressed as milligrams of QE equivalents per gram of extract (mg QE/g extract).

### 2.4. Determination of Total Phenolic Content

Total phenolic content (TPC) was analyzed using the Folin–Ciocalteu method [19]. First, 50 µL of each *B. hispida* extract (3 mg/mL in DI water) was mixed with 100 µL of 10% *v*/*v* Folin–Ciocalteu reagent. After 5 min, 50 µL of 10% *w*/*v* sodium carbonate was added. The mixture was vortexed and stored in the dark at ambient temperature for 2 h. Then, the absorbance of mixtures was measured using a UV spectrophotometer (UV-2600i, Shimadzu) at 765 nm against a blank without the extract. The TPC value was determined using a standard curve constructed with a gallic acid standard solution (0.0 to 1.0 mg/mL) and expressed as milligrams of gallic acid equivalents (GAEs) per gram of extract (mg GAE/g extract).

### 2.5. Antioxidant Activities

The *B. hispida* extracts were prepared to determine the antioxidant activities using various procedures. Trolox and ascorbic acid were used as standards in all assays.

#### 2.5.1. DPPH Radical Scavenging Assay

This experiment was determined using the modified method [20]. In total, 20 µL of the extract solution (0.0 to 1.0 mg/mL) was mixed with 180 µL of DPPH reagent (167 µM) and then incubated in the dark at ambient temperature for 30 min. The absorbance of the extract–reagent mixture was investigated at 520 nm using a microplate reader (SpectraMax M3, Molecular Devices, San Jose, CA, USA). The efficacy against DPPH radicals was expressed as the percentage of inhibition (% inhibition), which was estimated using the following equation:% Inhibition = [(Ac − Aw) − (Ae − Ab)/(Ac − Ab)] × 100(2)
where Ac is the absorbance of positive control that used ethanol mixed with tested reagent, Aw is the absorbance of negative control (ethanol), Ae is the absorbance of tested extract, and Ab is the absorbance of blank that used extract solution mixed with ethanol.

#### 2.5.2. ABTS Radical Scavenging Assay

The potential of *B. hispida* peel extracts was assessed in terms of scavenging activity using the ABTS assay following the method of Poomanee [21] with modifications. The ABTS reagent was prepared by mixing an ABTS solution in DI water (7 mM) with potassium persulfate solution in DI water (3 mM) in a ratio of 1:0.5 and incubated in a dark area for 18 h. The measurement was performed using a microplate reader (SpectraMax M3, Molecular Devices) at 734 nm. Before testing, the prepared ABTS reagent was diluted to provide an absorbance of 0.7 and was then used as a tested ABTS reagent. Altogether, 20 µL of the extract solution (0.0 to 1.0 mg/mL) and 180 µL of tested ABTS reagent were mixed and incubated in the dark at ambient temperature for 30 min. The absorbance of the extract–reagent mixture was measured at 520 nm using a microplate reader (SpectraMax M3, Molecular Devices). The % inhibition against ABTS radicals (% inhibition) was estimated using Equation (2) described above.

#### 2.5.3. Ferric Reducing Antioxidant Power (FRAP) Assay

This experiment was modified from the related study [22]. The tested samples, which were 1.0 mg/mL of each of the *B. hispida* extracts, were used in this experiment. Briefly, 20 µL of the extract solution was added to the mixed solution, which was 0.3 M acetate buffer (pH 3.6) with 10 mM TPTZ in 40 mM of 37% *w*/*v* hydrochloric solution. The extract–reagent mixture was incubated under dark conditions at ambient temperature for 5 min and then analyzed using a microplate reader (SpectraMax M3, Molecular Devices) at 595 nm. FRAP values are expressed as mg ferrous sulfate (FeSO_4_) per gram extract and were calculated from a linear regression equation constructed from various concentrations of FeSO_4_ solutions (100 to 1000 µM).

#### 2.5.4. Lipid Peroxidation Inhibition Assay

This investigation was based on the method in a related study [23]. The stock solution of *B. hispida* extracts was prepared at a concentration of 50.0 mg/mL in 50% DMSO. Then, 150 µL of extract solution was mixed with 100 µL of DI water, 350 µL of 20 mM phosphate buffer solution (PBS) (pH 7.0), 350 µL of 1.3% *v*/*v* linoleic acid in methanol and 50 µL of 25.14 mg/g 2,2′-azobis-(2-amidinopropane dihydrochloride) or (APPH) in 20 mM PBS (pH 7.0). The tested mixture was incubated at 50 °C for 4 h. Afterward, 5 µL of the mixture was added to a 96-well plate with 5 µL of 10% ammonium thiocyanate and 5 µL of 20 mM ferrous chloride, followed by adding 185 µL of 75% methanol. After 3 min, the absorbance of the mixed solution at 500 nm was determined using a Genios Pro microplate reader (Tecan, Crailsheim, Germany). The potential values expressed as % inhibition were calculated using Equation (2).

### 2.6. Anti-Aging Activities

#### 2.6.1. Collagenase Inhibition Assay

The collagenase inhibition activity of *B. hispida* extracts was determined following the protocol of Thring [24] with some modification. The tested collagenase enzyme at a concentration of 2.0 mg/mL was prepared at 25 °C in PBS pH 7.5, containing 50 mM Tricine, 10 mM calcium chloride and 400 mM sodium chloride. As a substrate, 1.0 mM of N-[3- (2 furyl) acryloyl]-Leu-Gly-Pro-Ala, also known as FALGPA, in PBS pH 7.5 was used. *B. hispida* extracts (10 µL) at a concentration of 0.5 mg/mL were added to a 96-well plate, and then the tested collagen enzyme (10 µL) was added, followed by the substrate (10 µL). The reaction mixture was incubated for 30 min at 37 °C. After that, this mixture was analyzed using a microplate reader (SPECTROstar Nano, BMG Labtech, Aylesbury, Buckinghamshire, UK) at a wavelength of 340 nm. For comparison, 0.5 mg/mL of EGCG standard solution was used as a positive control. The results expressed as % collagenase enzyme inhibition were calculated according to the following equation:% Collagenase enzyme inhibition = [(C − C0) − (Cs − Cb)/(C − C0)] × 100(3)
where C is the control absorbance (DI water mixed with collagen enzyme and substrate), C0 is the blank control absorbance (DI water mixed with PBS pH 7.5 and collagen enzyme), Cs is the tested extract absorbance and Cb is the extract blank absorbance (extract mixed with PBS and substrate).

#### 2.6.2. Hyaluronidase Inhibition Assay

This assay was performed to determine the potential of *B. hispida* extract against the hyaluronidase enzyme following the sigma protocol with slight modification [25]. The diluted hyaluronidase enzyme (2.0 mg/mL) was prepared at 25 °C in the enzymatic diluent, comprising a mixture of PBS pH 7.0, 77 mM sodium chloride and 0.01% BSA. Then, 0.03% hyaluronic acid mixed with PBS pH 5.3, namely PHS, was prepared and incubated at 80 °C. The *B. hispida* extract at a concentration of 0.5 mg/mL in DI water was used for analysis by mixing 50 µL of the *B. hispida* extract with 100 µL of diluted hyaluronidase enzyme solution and incubating the mixture at 37.5 °C for 10 min; then, 100 µL of 0.03% hyaluronic acid was added and continuously incubated at 37.5 °C for 45 min. After that, acetic albumin solution pH 3.75, containing 24 mM sodium acetate, 79 mM acetic acid and 0.1% BSA, was added and mixed at 25 °C. This mixture was subjected to analysis at 600 nm using a microplate reader (SPECTROstar Nano, BMG Labtech). Tannic acid, at the same concentration of the tested extract, was used as a positive control. The activity was shown as % hyaluronidase enzyme inhibition and calculated using the following equation:% Hyaluronidase enzyme inhibition = [(H − H0) − (Hs − Hb)/(H − Hb)] × 100(4)
where H is the control absorbance (tested mixture without the extract and using DI water as the tested sample), H0 is the blank control absorbance (DI water mixed with enzymatic diluent), Hs is the tested extract absorbance and Hb is the extract blank absorbance (extract mixed with PHS).

### 2.7. Chemical Marker Analysis by High-Performance Liquid Chromatography (HPLC)

Qualitative and quantitative analyses of the most promising *B. hispida* extract were performed using an HPLC system (Prominence LC2030C, Shimazu, Japan). A Knauer^®^ vertex III reversed-phase HPLC column C18 (250 mm × 20 mm) (KNAUER Wissenschaftliche Geräte GmbH, Berlin, Germany) was used as the stationary phase. A gradient system was used as an analytical method with a detector at a wavelength of 360 nm. The mobile phase consisted of A (0.1% acetic acid in water) and B (acetonitrile) with a flow rate of 1.0 mL/min for 33 min. Gradient elution of the mobile phase was carried out as follows: 0.01 min, 95% A; 2 min, 95% A; 7 min, 80% A; 22 min, 70% A; 23 min, 95% A; and 33 min, 95% A. The analytical samples were dissolved in absolute ethanol and filtered through a 0.45 μm filter (Whatman, Marlborough, MA, USA) before analysis with an injection volume of 10 μL. *B. hispida* extract at 1.0 mg/mL was prepared for analysis. Rutin was used as a reference compound to analyze the active compounds in *B. hispida* extract. The retention time of rutin was approximately 17 min. The concentration of rutin in *B. hispida* extract was calculated from the peak area using a linear equation constructed from a standard curve of rutin (0.0 to 50 µg/mL).

### 2.8. Molecular Docking Simulation

The main ingredients of *B. hispida* peel, including ascorbic acid, quercetin and rutin, were submitted to molecular docking simulation. The 3D chemical structures of these compounds were retrieved from the PubChem database (https://pubchem.ncbi.nlm.nih.gov/ accessed on 12 June 2023). All structures were geometrically optimized using Gaussian09w [26] with the HF/6-31G(d,p) [27] basis set. The enzymes involved in the aging process such as collagenase (PDB ID: 1CGL) [28] and hyaluronidase (PDB ID: 2PE4) [29] were obtained from the RCSB Protein Data Bank (PDB) database (https://www.rcsb.org/ accessed on 12 June 2023). All co-crystallized ligands and water molecules were removed from the structures. Before molecular docking, the Gasteiger’s and Kollman potential charges were assigned to ligands and proteins, respectively, using AutoDockTools 1.5.7 [30]. Molecular docking was carried out using the AutoDock Vina [31] program. The grid center of each protein was set according to its active site coordinates. The compound with the highest binding affinity was chosen as the representative structure for further analysis. Hydrogen bonds and hydrophobic interactions were identified using LigandScout 4.4.8 [32] and Discovery Studio Visualizer 2021 [33]. The molecular visualization was performed using the UCSF ChimeraX [34] Program.

### 2.9. Cell Cytotoxicity Test

To investigate the skin toxicity of *B. hispida* extracts, human dermal fibroblast (HDF) cells, obtained from the American Type Culture Collection (ATCC), were employed in this experiment. The test was performed using the MTT assay that was described in a related study with modifications [35]. The HDF cells (1 × 10^4^ cells/well) were cultured and treated in a 96-well plate with a medium culture containing 10% DMEM, 10% FBS and antibiotic solution (100 U/mL penicillin and 100 µg/mL streptomycin) and incubated at 37 °C in a 5% CO_2_ humidified atmosphere for 24 h. Then, the tested samples, *B*. *hispida* extracts at the concentration of 0.001 to 1.0 mg/mL, were added and continuously incubated in the same condition. After 24 h, the supernatant was removed, and 100 µL of 0.05 mg/mL MTT reagent was added to the well, followed by further incubation at 37 °C in a 5% CO_2_ humidified atmosphere for 6 h. The MTT reagent was then removed, and 100 µL of DMSO was added to dissolve the formazan crystals. After 15 min, the absorbance was measured at 560 nm using a microplate reader (Bio Tek Instruments, Winooski, VT, USA). The results were calculated and expressed as the percentage of cell viability using the following equation:% Cell viability = (A/A0) × 100(5)
where A is the absorbance of the tested sample and A0 is the absorbance of the control, namely the tested well without *B. hispida* extract.

### 2.10. In Vitro Irritation Test Using Hen’s Egg Chorioallantoic Membrane (HET-CAM) Assay

The irritation potential of *B. hispida* extracts was appraised using the HET-CAM assay according to a protocol described by Chaiyana [35]. This experiment serves as an alternative in vitro method for assessing skin irritation. For investigation using the HET-CAM assay, the placental membrane of a chicken embryo, providing a vascular network of capillaries, was used in the study. The fertilized hen eggs used in this study were obtained from the Faculty of Agriculture at Chiang Mai University, Thailand. Eggs aged between seven and nine days were employed. These eggs underwent incubation at a temperature of 37.5 ± 0.5 °C with a relative humidity of 62.5 ± 7.5%. The shell above the air cavitation of the egg was carefully removed using a rotating cutting blade attached to a Marathon champion 3 micromotor (Saeyang, Seoul, South Korea). A normal saline solution was then applied to an inner membrane that came into direct contact with the chorioallantoic membrane (CAM), followed by incubation under the same conditions as described above for 15 min. After this incubation period, the inner membrane was meticulously removed using forceps. Then, 30 µL of *B. hispida* extract at a concentration of 10 mg/mL was dropped onto the prepared CAM. In this study, the positive and negative controls were represented by 30 µL of 1% (*w*/*v*) sodium lauryl sulfate (SLS) and 30 µL of normal saline solution, respectively.

The assessment of irritation effects involved continuous observation using a stereomicroscope (Olympus, Tokyo, Japan) over a 5 min (300 s) period and expressed the short-term irritation effect. After 60 min of testing, observations were once again conducted to determine long-term irritation. The observed signs of irritation included vascular hemorrhage, vascular lysis and vascular irritation. The time of the first appearance of each of these irritating signs was recorded in seconds and was subsequently used to calculate an irritation score (IS) according to the following equation:IS = [((301 − Ht)/300) × 5] + [((301 − Lt)/300) × 7] + [((301 − Ct)/300) × 9](6)
where Ht is the time of the initial appearance of vascular hemorrhage, Lt is the time of the initial appearance of vascular lysis and Ct is the time point of the initial appearance of vascular coagulation. Based on the calculated IS, IS = 0.0 to 0.9 was classified as no irritation, IS = 1.0 to 4.9 was classified as mild irritation, IS = 5.0 to 8.9 was classified as moderate irritation and IS = 9.0 to 21.0 was classified as severe irritation.

### 2.11. Statistical Analysis

All measurements were performed independently in triplicate. The results are expressed as mean ± S.D. Statistical analysis was conducted using SPSS Software, Version 17.0 for Windows, and differences between groups were determined using one-way analysis of variance (ANOVA) followed by Tukey’s test. The differences were considered significant at a *p*-value ≤ 0.05.

## 3. Results

### 3.1. Plant Extraction

From the extraction using various solvents, the obtained extracts from *B. hispida* peels and the percentage yields of each extract are shown in Table 1. The highest extract yield was found for BW, followed by B95 and B50. The extracts appeared viscous and light brown and had specific odors.

### 3.2. Total Flavonoid and Total Phenolic Contents

The values of total flavonoid content (TFC) and total phenolic content (TPC) in *B. hispida* extracts are shown in Table 2. Flavonoid contents in different extracts were calculated from the regression equation of the quercetin standard curve (y = 0.3502x + 0.0565, r^2^ = 0.993) and expressed as mg QE/g extract. Phenolic contents in different extracts were calculated from the regression equation of the gallic acid standard curve (y = 8.2221x + 0.006, r^2^ = 0.998) and expressed as mg GAE/g extract.

The highest TFC was found in B95 with a value of 212.88 ± 4.73 mg QE/g extract and was followed by the TFCs of BW and B50, consecutively. However, the highest TPC was found in B50 with the value of 140.43 ± 0.77 mg GAE/g extract and was followed by the TPCs of B95 and BW, consecutively. These results might be related to the extracted solvents; some flavonoid and phenolic compounds are well extracted in semi-polar solvents such as 95% ethanol, which was presented in the study as having a high value of TFC and TPC in B95.

### 3.3. Antioxidant Activities

The antioxidative activities of *B. hispida* peel extracts were investigated using various methods to evaluate various mechanisms of action. The results of *B. hispida* extracts from these procedures are shown in Table 3.

DPPH and ABTS assays were performed to determine the radical scavenging activity of the extracts. From the DPPH assay, the results were expressed as the concentration of extract that could be active against radicals where the % inhibition is equal to 50 (IC_50_). The IC_50_ values of extracts in the DPPH assay were calculated from the regression equations of each extract, namely y = 0.0128x + 12.75, r^2^ = 0.986 for B95; y = 0.0308x + 2.6124, r^2^ = 0.9895 for B50; and y = 1.3828x + 16.589, r^2^ = 0.925 for BW. The standard compounds that were considered in comparison were ascorbic acid and Trolox. The IC_50_ values of these standard compounds were also calculated from the regression equations of each compound, namely y = 6.1021 + 20.349, r^2^ = 0.936 for ascorbic acid and y = 3.031 + 14.631, r^2^ = 0.989 for Trolox. The results demonstrated that B50 exhibited the significantly highest scavenging activity with an IC_50_ value of 1.73 ± 0.13 mg/mL, followed by B95 with an IC_50_ of 2.91 ± 0.07 mg/mL and BW with an IC_50_ value of 23.63 ± 0.60 mg/mL (*p* < 0.05).

The ABTS assay results were expressed as IC_50_ values, the same as the results obtained from the DPPH assay. IC_50_ values of the *B. hispida* extracts were calculated using the equations of linear regression of each extract. The obtained equations used to calculate IC_50_ values of B95, B50 and BW were y = 0.227x + 9.7036 (r^2^ = 0.9907), y = 0.2387x + 10.612 (r^2^ = 0.9708) and y = 0.1387x + 10.685 (r^2^ = 0.9279), respectively. Ascorbic acid and Trolox were used as the standard compounds, and IC_50_ values were also calculated from the regression equations. The calculated equations of ascorbic acid and Trolox were y = 7.2955x + 6.7331 (r^2^ = 0.9841) and y = 3.9231x + 2.1272 (r^2^ = 0.9803), respectively. The obtained IC_50_ values from ABTS assays showed that B95 and B50 provided the highest ABTS scavenging activity with IC_50_ values of 0.17± 3.13 mg/mL and 0.16 ± 3.95 mg/mL, followed by BW with an IC_50_ value of 0.27 ± 12.09 mg/mL (*p* < 0.05).

The FRAP assay was used to determine the ability of radical reduction. The results were expressed as the mg FeSO_4_/g extract which was calculated from the regression equation of FeSO_4_ (y = 6.724x + 0.1818, r^2^ = 0.995). A high FRAP value indicated a high radical reduction activity. *B. hispida* extracts (B95, B50 and BW) at a concentration of 1.0 mg/mL were used in this experiment. The BW showed the highest reducing property with a FRAP value of 17.33 ± 0.19 mg FeSO_4_/g extract. Nevertheless, B95 showed less activity in comparison with others (*p* < 0.05).

The lipid peroxidation inhibition assay was performed to determine the antioxidant ability against oxidative degradation of lipids by peroxide radicals. The results were expressed as % inhibition obtained from B95, B50 and BW extracts including the standard compounds (ascorbic acid and Trolox) at the same concentration of 0.1 mg/mL. B95 possessed high activity with the % inhibition of 53.42 ± 0.02, against peroxide radicals, followed by B50 and BW, consecutively.

Ascorbic acid and Trolox, the compounds that were used for comparing the antioxidant activity in all tested assays, showed the significantly strongest activities when compared with all *B. hispida* peel extracts (*p* < 0.05). Interestingly, the overall antioxidative assays illustrated that B95 obtained from *B. hispida* peel has potential antioxidant activity related to all antioxidant mechanisms.

### 3.4. Anti-Aging Activities

To determine anti-aging activities of *B. hispida* peel extracts, the ability to inhibit collagenase and hyaluronidase enzymes was investigated. The results were expressed as % collagenase enzyme inhibition and % hyaluronidase enzyme inhibition. The obtained inhibitory effects of *B. hispida* peel extracts are illustrated in Figure 2. Among the samples tested against the collagenase enzyme (Figure 2A), B95 showed an inhibition effect that does not significantly differ (*p* > 0.05) from EGCG, a standard compound, at the same concentration of 0.5 mg/mL. B95 showed an inhibition of the collagenase enzyme with the value of 41.68 ± 0.92%. Furthermore, the % collagenase inhibition exhibited by B50 and BW did not demonstrate statistically significant differences when compared with that of B95 (*p* > 0.05). Concerning the activity against the hyaluronidase enzyme, the obtained results are demonstrated in Figure 2B. B95, with a value of 29.17 ± 0.66%, showed obviously higher hyaluronidase inhibition activity than other extracts (*p* < 0.05). Nevertheless, B50 and BW demonstrated hyaluronidase inhibition of the same potential (*p* > 0.05). However, all *B*. *hispida* peel extracts possessed an inhibitory effect against the hyaluronidase enzyme that was less than that of tannic acid, the standard compound in this experiment (*p* < 0.05).

### 3.5. Chemical Marker Analysis by High-Performance Liquid Chromatography (HPLC)

The results obtained from antioxidant and anti-aging assays showed that B95 exhibited the most potent effects among all the tested extracts of *B. hispida*. Consequently, B95 was selected to determine its chemical marker using HPLC. Rutin, a reference flavonoid compound used to identify the active compound in B95, was also included in the analysis for comparison. The HPLC chromatograms of B95 and rutin are presented in Figure 3. Several components were detected in B95, as indicated by the observed HPLC chromatograms; one of these components was rutin, which exhibited a retention time of approximately 17 min. The quantification of rutin in B95 was calculated using the linear equation of rutin, namely y = 8914.9x + 793.63 (r^2^ = 0.9999). The results demonstrated that 1 mg of B95 contained 4.81 ± 0.03 µg of rutin.

### 3.6. Molecular Docking Simulation

The main ingredients of *B. hispida* peel extract, including ascorbic acid, quercetin and rutin, were submitted to molecular docking simulation. The molecular docking results indicated that quercetin exhibited the highest binding affinity against the collagenase enzyme (z8.9 kcal/mol), followed by rutin (−7.9 kcal/mol) and ascorbic acid (−6.3 kcal/mol), consecutively. The molecular recognition of these compounds against collagenase is depicted in Figure 4. All compounds fit well in the binding pocket of collagenase, as shown in Figure 4A. Ascorbic acid interacts with key amino acids, including Ala182, Arg214, His218, Glu219 and Pro238, via hydrogen bonding and also forms a metal interaction with Zn301 (Figure 4B). Quercetin forms hydrogen bonds with Arg214, Glu219 and Tyr237 and interacts with Leu181 and Val215 via hydrophobic interactions (Figure 4C). Rutin exhibits interactions with amino acids such as Asn180, Arg214, Tyr237 and Tyr240 through hydrogen bonds and forms hydrophobic interactions with Val215 (Figure 4D).

The molecular docking results for the hyaluronidase enzyme showed that rutin possessed the highest docking affinity (−9.8 kcal/mol), followed by quercetin (−8.1 kcal/mol) and ascorbic acid (−5.6 kcal/mol). The graphical illustration indicates that the molecular binding patterns of the compounds differ in terms of their positions within the binding pocket (Figure 4E). Ascorbic acid fits well in the narrow pocket to form hydrogen bonds with residues including Tyr75, Asp129, Glu131 and Tyr286 (Figure 4F). Quercetin interacts with Trp130, Glu131, Asp206, Tyr210, Ser245 and Asp292 via hydrogen bonds and forms hydrophobic interactions with Tyr202 and Phe204 (Figure 4G). Rutin is located on the same site as quercetin. The core structure of rutin forms hydrogen bonds with Glu131, Gly203, Tyr202, Asp206, Ser245 and Asp292, and the glycoside substituent forms hydrogen bonds with Arg134, Trp141, Tyr247 and Tyr261 (Figure 3).

### 3.7. Cell Cytotoxicity Test

A cytotoxicity test of B. hispida extracts (B95, B50 and BW) on HAF cells was performed using the MTT assay. The tested samples showing cell viability of above 80% were regarded as nontoxic to the cells. The obtained results after 24 h of exposure of the tested samples to the HAF cells are shown in Figure 5. All B. hispida extracts were not toxic to HAF cells at concentrations of 0.001 to 0.01 mg/mL, with cell viability of more than 90%. In addition, B50 and BW at a concentration of 1.0 mg/mL also exhibited nontoxicity to HAF cells with a % cell viability of 99.74 ± 8.63, and 92.35 ± 17.30, respectively. However, B95 at a concentration of 1.0 mg/mL showed slight toxicity to HAF cells with a % cell viability of 77.58 ± 2.82%.

### 3.8. In Vitro Irritation Test Using the HET-CAM Assay

To further investigate the effective activities of *B. hispida* peel extracts in vivo and in a clinical study, an in vitro irritation test using the HET-CAM assay was then employed. The irritation induced by *B. hispida* peel extracts (B95, B50 and BW) was evaluated and compared with that of the negative control, 0.9% *w*/*v* NaCl, and the positive control, 1% (*w*/*v*) SLS. The obtained results as irritation scores are shown in Table 4, and the observations of the alterations under a stereomicroscope are shown in Figure 6. None of the *B. hispida* peel extracts at a concentration of 10 mg/mL induced considerable alterations, namely vascular hemorrhage, vascular lysis and vascular irritation, in short-term testing (5 min) and long-term testing (60 min), similar to the result obtained from 0.9% *w*/*v* NaCl. However, 1% (*w*/*v*) SLS, the positive control, exhibited signs of severe irritation in 5 min of testing with vascular hemorrhages and vascular lysis.

## 4. Discussion

It has been realized that the potential of antioxidants against free radical damage to the skin is a significant property that can decelerate skin aging and skin damage related to transduction pathways and epigenetic changes caused by oxidative stress [36]. Moreover, countless studies have explored the antioxidant activity from novel natural sources, including plants, animals, insects, algae and waste products, to protect the skin from damage caused by free radicals [37,38]. Phenolic compounds are powerful antioxidant agents with wide mechanisms of action including the capacity to scavenge free radicals such as hydroxyl radical (^•^OH) and superoxide anion (O_2_^−•^) and chelate metal ions (Fe^2+^, Fe^3+^, Cu^2+^ and Cu^+^) which react to hydrogen peroxide (H_2_O_2_). Highly reactive ^▪^OH is formed by the dismutation of O2^−•^ by SOD, enhancing the activity or expression of intracellular antioxidant enzymes [39]. Flavonoids are a group of natural substances with variable phenolic structures. Flavones and catechins have been reported to be the most powerful flavonoids for protecting the body against ROS [40].

*B. hispida* is a potentially edible plant that has been used for remedies and protection against many diseases related to the phytochemicals in the used part [11]. The obtained extracts of *B. hispida* in this study showed a high yield in BW and B95, which could indicate that the chemical constituents in *B. hispida* peel are well solubilized in water and 95% ethanol. Related studies have reported the presence of phenols, alkaloids, saponins, steroids, carbohydrates and flavonoids in chloroform, alcoholic and aqueous extracts obtained from *B. hispida* peel [10,13]. This was consistent with the results of this study that found flavonoids in *B. hispida* peel extracts, especially in B95 (95% *v*/*v* ethanolic extract). Moreover, phenolic compounds were found in all extracts. The phytochemical profile of phenolic compounds in *B. hispida* has been reported by Islam et al., and the results revealed the presence of many bioactive chemicals, such as quercetin, rutin, astilbin, catechin, naringenin and hispidulin [11]. The antioxidant activities of *B. hispida* extracts were investigated with standard procedures to evaluate the different mechanisms of action. DPPH and ABTS methods have commonly been used to evaluate the antioxidant activity of compounds that act as scavengers of O2^−•^ and ^•^OH radicals [41]. The FRAP assay is a method that possesses the power of antioxidation by reducing ferric iron (Fe^3+^) to ferrous iron (Fe^2+^) [42]. In this study, B95 obviously possessed antioxidant activities with free radical scavenging mechanisms via DPPH and ABTS assays. Moreover, it could inhibit lipid peroxidation by stopping lipid chain reactions. Further, BW possessed good potential to reduce Fe^3+^ via the FRAP assay. These results are consistent with related studies reporting that the methanolic extracts of *B. hispida* peel provided antioxidant activity against free radicals when determined using DPPH and FRAP methods [10,43]. In addition, the high flavonoid and phenolic contents found in B95 possessed good potential antioxidant activities in all assays, referring to the synergistic action of both compounds [44].

Collagenase is a key enzyme that acts by degrading collagen in the extracellular matrix, promoting premature skin aging. Hyaluronidase is also a key enzyme that degrades hyaluronic acid, a critical component of the extracellular matrix in the skin for maintaining the normal hydration of the skin [45,46]. The inhibition of both enzymes provides the potential to prevent skin aging and its signs. Our study found that the strongest anticollagenase activity was provided by B95 (72.70 ± 1.24%), and the strongest antihyaluronidase activity was also exhibited by B95 (74.88 ± 4.84%). The results presumably recommend that flavonoid compounds that were found to possess the highest B95 levels may be the active compounds for these activities. In a related study, rutin and quercetin, which belong to the group of flavonoid compounds, were detected in the fruit of *B*. *hispida*, indicating good anti-aging activities [16,47]. In addition, the literature extensively supports the efficacy of flavonols as inhibitors of collagenase [48]. Quercetin and kaempferol exhibited a more significant inhibitory effect compared with flavones, isoflavones and flavanones, with the latter demonstrating a very negligible impact. The arrangement of hydroxyl groups in the B-ring of the flavonoid structure may play a crucial role in determining the inhibitory effect on enzyme activity [49]. Similarly, the antioxidative and anti-enzymatic activities of Thai plants were found to be influenced by their phenolic and flavonoid contents [6]. Thus, both compounds were investigated in a molecular docking simulation against collagenase and hyaluronidase enzymes.

Various phytochemicals have been identified in *B. hispida* extracts, primarily in the fruit portion [16]. However, limited information is available regarding the composition of phytochemicals obtained from *B. hispida* peel. In this study, the most promising *B. hispida* extract, namely B95, was used to determine the phytochemical active compounds through HPLC analysis. Rutin, a flavonoid that has been detected in various parts of *B. hispida*, was employed as a reference compound for the HPLC analysis of B95. This analysis revealed the presence of numerous compounds, which manifested as multiple HPLC peaks observed in the chromatogram. Rutin is one of the compounds identified in B95. This result aligns with related studies that have also detected rutin in other parts of *B. hispida* [50,51]. However, it is necessary to identify and characterize the other compounds in B95. This will provide a comprehensive understanding of B95′s composition and support the feasibility of its mechanism of action and efficacy in various applications. Conducting a broader analysis to identify and understand the various compounds in B95 can significantly enhance the ability to assess its potential benefits and applications. Molecular docking contributes to a better understanding of the molecular basis underlying the interaction between promising compounds from *B. hispida* and the collagenase and hyaluronidase enzymes, which are the key enzymes related to anti-aging properties. Ascorbic acid, quercetin and rutin were identified as the main bioactive compounds in the *B. hispida* peel extract and were further investigated for their molecular recognition within the binding sites of these enzymes. All phytochemicals are capable of binding to the collagenase active site cleft through hydrogen and hydrophobic interactions. The strength of the binding of these phytochemicals correlates to the number of hydrogen bonds and steric effects. Quercetin and rutin form interactions with Arg214 at the bottom of the S1′ pocket, which is suitable in size for an aromatic ring [52]. Moreover, the hydrogen bond with Glu219 also supports the binding affinity of quercetin within the binding pocket [28]. In hyaluronidase, rutin and quercetin form hydrogen bonds with Asp129 and Glu131, which are essential amino acids for the catalytic site of hyaluronidase [53,54]. The interaction of quercetin and rutin with Tyr202 has been observed, further supporting the binding strength [55]. This interaction information can provide insights into the potential use of the phytochemicals from *B. hispida* peel extract as anti-aging agents that inhibit the activity of collagenase and hyaluronidase, which are enzymes associated with the skin aging processes of collagen degradation and hyaluronic acid breakdown.

Regarding safety, the cytotoxicity and the irritation of *B. hispida* peel extracts need to be investigated before application and further development for food, cosmetics or cosmeceutical products. The cytotoxicity of all *B. hispida* peel extracts on human dermal fibroblast cells was evaluated using the MTT assay, which is a standard protocol for assessing in vitro cytotoxicity. This assay is based on the reduction of the yellow tetrazolium salt of MTT to purple formazan crystals by metabolically active cells. Thus, the high detection of purple formazan crystals in the experiment indicates a high density of living cells. A test sample showing cell viability above 80% is considered as safe or possessing non-cytotoxicity [56,57]. In the present study, *B. hispida* peel extracts (B95, B50 and BW) demonstrated cell viability of more than 80% after being exposed to the cells for 24 h. These results were consistent with one related study reporting that *B. hispida* extract is not toxic to normal cells [58]. Therefore, it could be concluded that *B. hispida* peel extracts are not harmful to dermal cells and can be safely applied on the skin.

As the irritation test, the HET-CAM assay was used to assess the irritation properties of *B. hispida* peel extracts. This protocol was developed by Lüepke [59] and is employed as an alternative to the Draize test for screening the potential of irritation and anti-irritation effect of cosmetic formulations and ingredients [60]. The HET-CAM test was compared with results obtained from the irritation test with human skin, and the results led to the conclusion that significant comparable efficacy patterns were found [61]. Thus, the HET-CAM assay has been used as a preliminary assessment of irritation before clinical trials. The results obtained in our study revealed that none of the *B. hispida* peel extracts showed any signs of irritation on the chorioallantoic membrane. The signs of irritation include hemorrhage, coagulation and vascular lysis. These results revealed that *B. hispida* peel extracts are suitable for further developing topical products or skin care products with safety similar to that of other extracts that have been reported to have no irritation potential after being tested using the HET-CAM assay [7,31,62]

## 5. Conclusions

In the present study, we have demonstrated that *B. hispida* peel extracts received from postconsumer waste, containing flavonoid and phenolic compounds, constitute a potential natural source. All the extracts possessed significant antioxidant and anti-aging activities, especially activities against enzymes that cause skin aging. The extract obtained from 95% ethanol as the extraction solvent showed the most promising antioxidant activities involving the mechanisms of radical scavenging of DPPH and ABTS radicals, reduced free radicals in the FRAP assay and inhibited peroxide radicals. Regarding the investigation of anti-aging activities, *B. hispida* peel extract, obtained from 95% ethanol, possesses good ability to inhibit the collagenase enzyme at the same potential as epigallocatechin gallate (standard control). Additionally, this extract also exhibits an inhibitory effect on the hyaluronidase enzyme. Based on the molecular docking results, the interaction of *B. hispida* peel extracts with collagenase and hyaluronidase suggests that they have the potential to be used as anti-aging agents. The in vitro toxicity against human dermal fibroblast cells and the in vitro irritation assay using hen’s egg chorioallantoic membrane demonstrate that all *B. hispida* peel extracts possess no toxicity and produce no irritating effect. In conclusion, these valuable results provide scientific evidence to support further studies concerning the physiochemical properties of *B. hispida* extracts and their active compounds. This research serves as a preliminary study via in vitro investigation; the mechanisms of in vivo antioxidant and in vivo anti-aging effects could be clarified in further investigations. Additionally, developing products containing *B. hispida* extracts for use in food, cosmetics or cosmeceuticals, as well as clinical studies, could also be investigated. Expanding upon this research can open opportunities for a wide range of practical applications.

## Figures and Tables

**Figure 1 foods-12-03555-f001:**
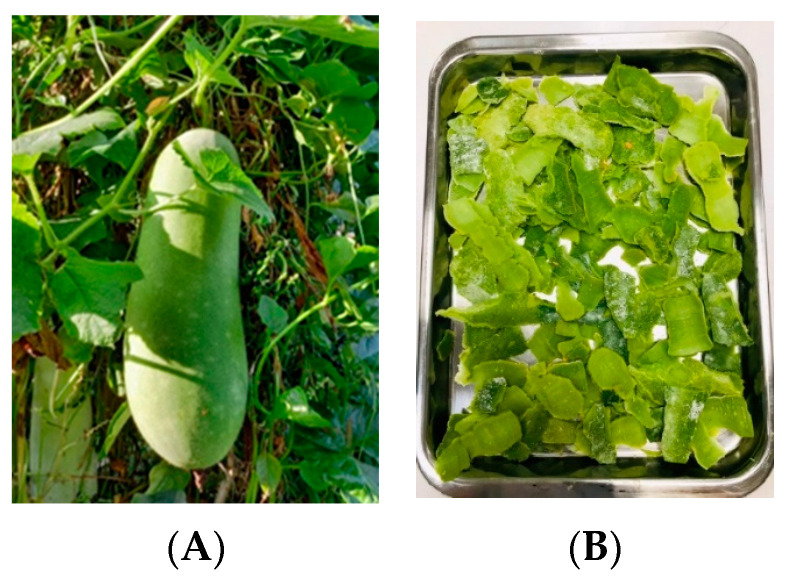
*B. hispida fruit* (**A**) and its peel (**B**) used in investigation.

**Figure 2 foods-12-03555-f002:**
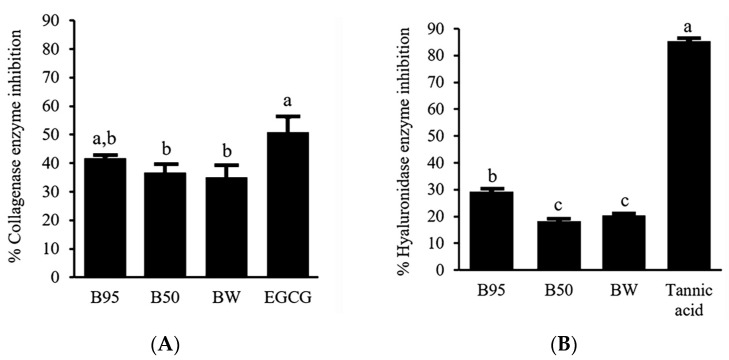
Potential of *B. hispida* extracts against the collagenase enzyme compared with EGCG (**A**) and the hyaluronidase enzyme compared with tannic acid (**B**). Different letters indicate significance level at *p* < 0.05.

**Figure 3 foods-12-03555-f003:**
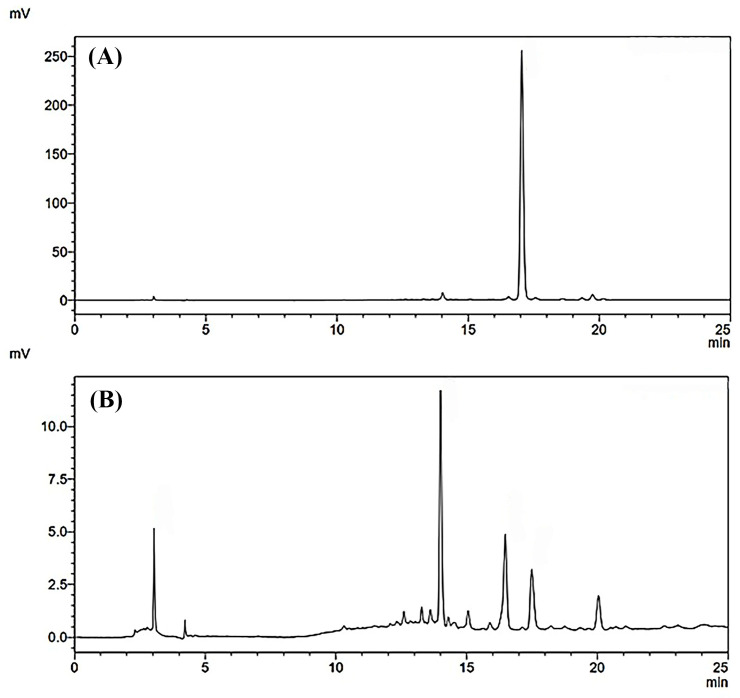
HPLC chromatograms of rutin at the concentration of 50 µg/mL (**A**) and the most promising *B. hispida* extract, B95 (**B**) at a concentration of 1.0 mg/mL, analysis at 360 nm.

**Figure 4 foods-12-03555-f004:**
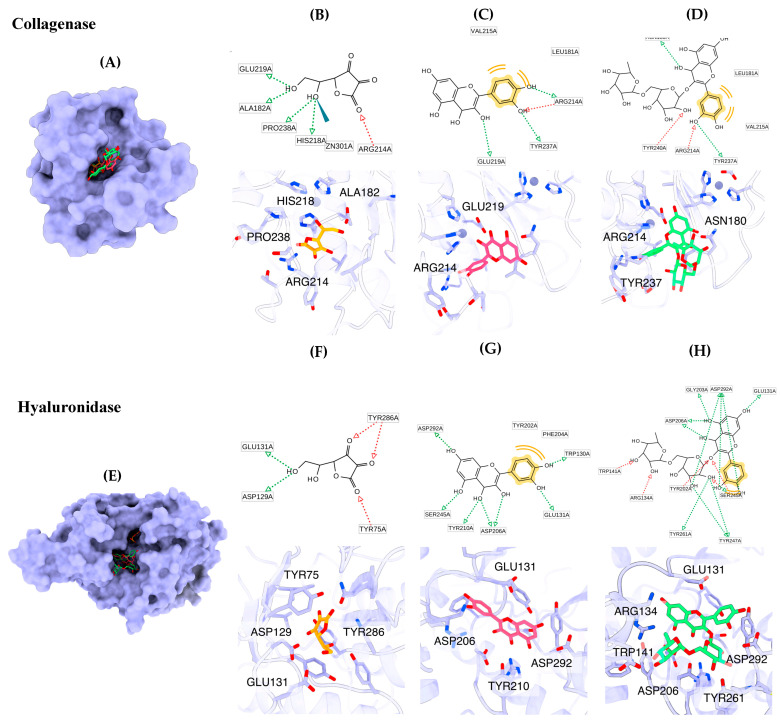
Binding modes of representative phytochemicals binding to collagenase (**A**) and hyaluronidase (**E**). The 2D and 3D interactions of ascorbic acid (**B**), quercetin (**C**) and rutin (**D**) complexed with collagenase, along with the 2D and 3D binding modes of ascorbic acid (**F**), quercetin (**G**) and rutin (**H**) in the binding site of hyaluronidase. The pharmacophore features include hydrogen bond acceptor (red arrow), hydrogen bond donor (green arrow) and hydrophobic property (yellow).

**Figure 5 foods-12-03555-f005:**
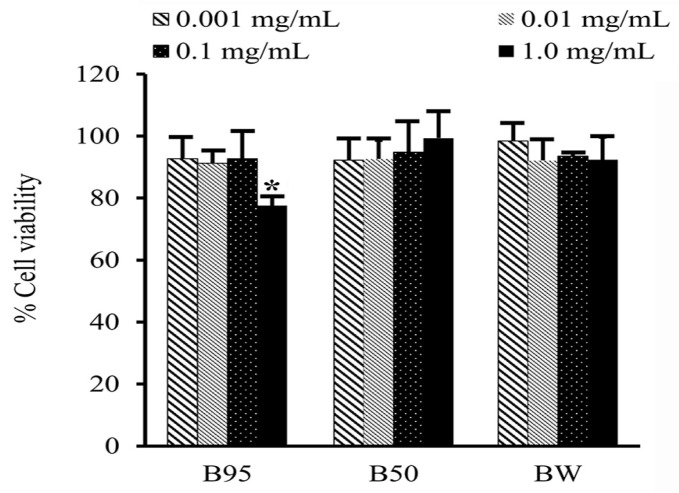
Percentage cell viability of each *B. hispida* extract at concentrations from 0.001 to 1.0 mg/mL. The values are presented mean ± S.D. (*n* = 3). Asterisk (*) represents a different value at *p* < 0.05.

**Figure 6 foods-12-03555-f006:**
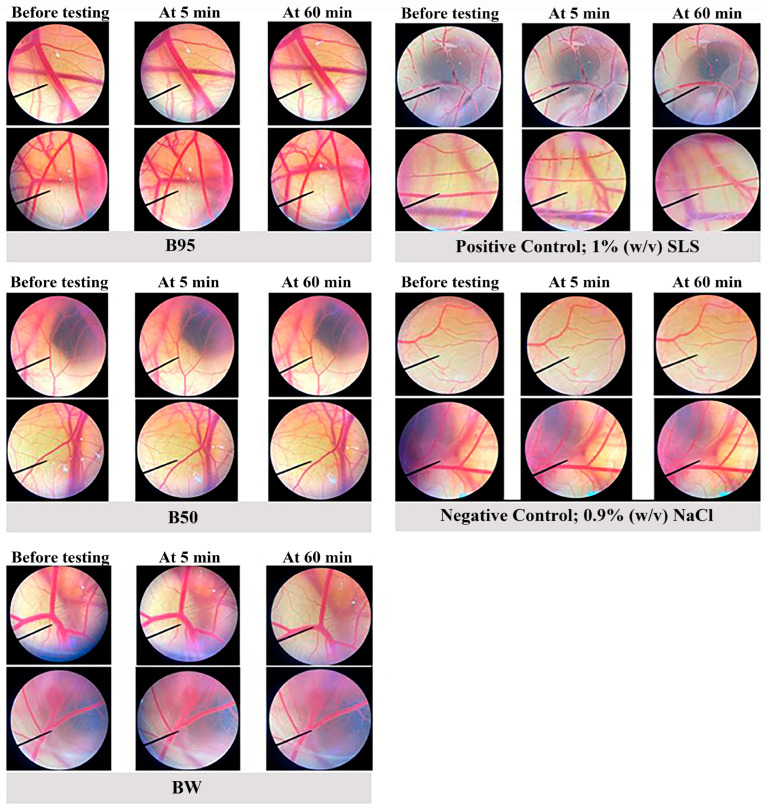
The stereomicroscope images of CAM vasculature before testing, at 5 min after treatment and at 60 min after treatment with 10 mg/mL of each extract (B95, B50 and BW), 0.9% *w*/*v* NaCl and 1% *w*/*v* SLS.

**Table 1 foods-12-03555-t001:** Percentage yields of *B. hispida* peel extracts from various solvents.

Extract	Solvent	Yield (%)
B95	95% ethanol	29.6 ± 1.28 ^b^
B50	50% ethanol	15.2 ± 0.98 ^c^
BW	DI water	32.7 ± 0.87 ^a^

The results show the average yield values of extracts ± S.D., and the different letters indicate significance level at *p* < 0.05.

**Table 2 foods-12-03555-t002:** Total phenolic and total flavonoid contents of *B. hispida* peel extracts.

Extract	Total Flavonoid Content (mg QE/g Extract)	Total Phenolic Content (mg GAE/g Extract)
B95	212.88 ± 4.73 ^a^	131.52 ± 0.38 ^b^
B50	54.18 ± 0.11 ^c^	140.43 ± 0.77 ^a^
BW	60.83 ± 0.86 ^b^	115.91 ± 0.78 ^c^

The results show the average values ± S.D. Different letters indicate significance level at *p* < 0.05.

**Table 3 foods-12-03555-t003:** Antioxidant activities with various assays.

Tested Sample	DPPH Assay * (IC_50_; mg/mL)	ABTS Assay * (IC_50_; mg/mL)	FRAP Assay * (mg FeSO_4_/g Extract)	Lipid Peroxidation Assay * (% Inhibition)
B95	2.91 ± 0.07 ^c^	0.17± 3.13 ^c^	0.75 ± 0.09 ^e^	53.42 ± 0.02 ^c^
B50	1.73 ± 0.13 ^b^	0.16 ± 3.95 ^c^	2.46 ± 0.18 ^d^	40.31 ± 0.01 ^e^
BW	23.63 ± 0.60 ^d^	0.27 ± 12.09 ^d^	17.33 ± 0.19 ^c^	45.39 ± 0.86 ^d^
Ascorbic acid	0.01 ± 0.04 ^a^	0.00002 ± 0.02 ^a^	1926.16 ± 21.40 ^a^	94.17 ± 3.74 ^a^
Trolox	0.013 ± 0.26 ^a^	0.00544 ± 1.71 ^b^	566.65 ± 27.41 ^b^	78.82 ± 3.36 ^b^

* Results show the average antioxidant values of extracts ± S.D. from different assays. Different letters indicate significance level at *p* < 0.05.

**Table 4 foods-12-03555-t004:** Irritation score of *B. hispida* peel extracts, 0.9% *w*/*v* NaCl and 1% *w*/*v* SLS.

Tested Compound	Irritation Score *	Irritation Category
B95	0.0 ± 0.0 ^b^	No irritation
B50	0.0 ± 0.0 ^b^	No irritation
BW	0.0 ± 0.0 ^b^	No irritation
0.9% (*w*/*v*) NaCl	0.0 ± 0.0 ^b^	No irritation
1% (*w*/*v*) SLS	11.1 ± 0.5 ^a^	Severe irritation

* The results are shown as the average IS ± S.D. Different letters indicate significance level at *p* < 0.05.

## Data Availability

The data presented in this study are available from the corresponding author upon reasonable request.

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
