# Peer review of "Influence of Benincasa hispida Peel Extracts on Antioxidant and Anti-Aging Activities, including Molecular Docking Simulation"

_foods, 2023, doi:10.3390/foods12193555_

Round 1

Reviewer 1 Report

The study “The Influence of Benincasa hispida Peel Extracts on Antioxidant and Anti-aging Activities, Including Molecular Docking Simulation” is well-written and can be accepted after revising following comments.

Comment 1: English write-up of the article must be improved by native speaker

Comment 2: There must be problem statement of article mentioned in the introduction part, which is lacking, indicating the importance of this study

Comment 3: Where is the practical significance of this study

Comment 4: Why were the Benincasa hispida (Thunb.) chosen besides that there is multiple option with same characteristics

Comment 5: The material and method section are well-explained

Comment 6: Section 3.1 and 3.2: need to elaborate the results

Comment 7: Line 327: compound which were y = 6.1021 “replace with” compound, which were y = 6.1021

Comment 8: Line 451: scavenge free radicals such as hydroxyl radical “replace with” scavenge free radicals such as, hydroxyl radical

Comment 9: Need to improve the minor English Grammer mistakes

Comment 10: Overall, the results and discussion part are well-explained

Comment 11: Write down the limitation, future work and recommendation for next study

Moderate English changes required

Author Response

Thank you very much for taking the time to review our manuscript. Please find the detailed responses in the attached file and the corresponding revisions/corrections highlighted in the re-submitted files.

Comments

Responses

English write-up of the article must be improved by native speaker

It already checks following the comment from reviewer.

There must be problem statement of article mentioned in the introduction part, which is lacking, indicating the importance of this study

More information about the problem statement of article was added in Line 63-67 following the comment from reviewer.

Where is the practical significance of this study

The emphasis of this  study was added in the conclusion part.

Why were the Benincasa hispida (Thunb.) chosen besides that there is multiple option with same characteristics

Because B. hispida is trended for high consumption in Thailand, especially in Northern part which lead to have a wasted product (peel) from consumer and industry.

Section 3.1 and 3.2: need to elaborate the results

It has been adjusted following  the comment from reviewer that present in Line 340-342.

Line 327: compound which were y = 6.1021 “replace with” compound, which were y = 6.1021

It has been adjusted following  the comment from reviewer.

Line 451: scavenge free radicals such as hydroxyl radical “replace with” scavenge free radicals such as, hydroxyl radical

It has been adjusted following  the comment from reviewer

Need to improve the minor English Grammer mistakes

It already checks following the comment from reviewer

Write down the limitation, future work and recommendation for next study

The limitation and recommendation are added in Line 623-628.

Reviewer 2 Report

Why was 50 oC used to dry the shell of B. hispida? Please give reasons, if there is previous research or preliminary experiments, they should be referenced.

Please specify the type and manufacturer of the drying equipment.

Drying for 24-48 hours is a long time interval, clarification is necessary, based on the description, the experiment cannot be repeated.

How long did the vacuum evaporation of the extracts take and at what temperature?

Samples B95 and B50 were prepared by vacuum evaporation, DI by lyophilization. Explain why the results are considered comparable despite the different methods.

 Why was ethanol chosen as a solvent next to water? On what basis did they decide the 95 and 50%? From an economic point of view, how do you judge the concentration of solvents? Is it possible that 75% would be enough?

2.5.3. In what units were the FRAP results given?

2.5.4. What does the abbreviation APPH mean?

8. page, 353. és 354. lines: It was found that the B95 possessed the highest activity with the percentage inhibition of 53.42 ± 0.02, against peroxide radicals followed by the B50 and BW, respectively.

BW and B50 are the correct order.

B95 extract was the best for inhibiting collagenase and hyaluronidase, even though it ranked first in various antioxidant assays only for ABTS and lipid perixodation assay. In terms of DPPH, it ranks second and last in terms of FRAP.

What mechanisms could be behind the fact that, despite the results, the antioxidant and anti-aging activities are still promising?

Author Response

Thank you very much for taking the time to review our manuscript. Please find the detailed responses in the attached file and the corresponding revisions/corrections highlighted in the re-submitted files.

Comments

Responses

Why was 50 oC used to dry the shell of B. hispida? Please give reasons, if there is previous research or preliminary experiments, they should be referenced.

Drying plant material at 50°C for extraction is a common practice in many botanical and phytochemical studies. The choice of temperature for drying plant material depends on several factors, and 50°C is often chosen for specific reasons:                                                                     

1)      Preservation of bioactive compounds: Drying plant material at moderate temperatures like 50°C helps preserve heat-sensitive bioactive compounds, including volatile organic compounds (VOCs), essential oils, and phytochemicals. These compounds can be easily degraded or lost at higher drying temperatures.

2)      Minimization of enzyme activity: Plant tissues contain enzymes that can break down bioactive compounds and contribute to degradation. Drying at 50°C effectively deactivates enzymes, slowing down their activity and preserving the integrity of the compounds of interest. 

3)      Balanced drying speed: Drying at 50°C strikes a balance between drying speed and preservation. While lower temperatures can preserve compounds better, they also slow down the drying process significantly. Higher temperatures can speed up drying but may lead to more compound degradation.                                 

4)      Safety and energy efficiency: Drying at 50°C is generally considered safe and energy-efficient for many types of plant materials. It's not too high a temperature to pose a fire hazard, and it doesn't require excessive energy consumption.     

It's important to note that the choice of drying temperature can vary depending on the specific plant material, its intended use, and the compounds you want to extract. Some plant materials may require even lower drying temperatures to preserve their properties effectively, while others may tolerate slightly higher temperatures. Therefore, in this study, the peel was dried at 50°C for extraction for the reasons stated above, and Bellur et al. (2015) also reported on the preparation material of cucurbitaceous vegetables (Benincasa hispida, Luffa acutangular, and Sechium edule) by drying the peel in an oven at 50±1°C before determining their chemical composition.

Reference

Nagarajaiah, S.B.; Prakash, J. Chemical composition and bioactive potential of dehydrated peels of benincasa hispida, luffa acutangula, and sechium edule. Journal of Herbs, Spices and Medicinal Plants 2015, 21, 193-202.

Please specify the type and manufacturer of the drying equipment.

Following the comments from reviewer, the hot air oven belongs to Memmert Ltd, Wisconsin USA which was added in Line 102.

Drying for 24-48 hours is a long time interval, clarification is necessary, based on the description, the experiment cannot be repeated.

According to Bellur et al. (2015), the peel of B. hispida possessed a high percentage of moisture content (94.10 ± 0.14% or 6.08 ± 0.14g/100g of fresh peel) and a water absorption capacity was 680.0 0.00 mL/100g of fresh peel weight. Based on these findings, we chose to keep drying the peel at 50°C for 48 h, or until its ultimate weight remained constant. Furthermore, Kumar et al. (2012) proposed that B. hispida peels should be cut into tiny pieces and dried as a powder under vacuum at 60°C for 48 h before exposure to sensitive extraction.

Reference

Chidan Kumar, C.S.; Mythilj, R.; Chandraju, S. Extraction and mass characterization of sugars from Ash gourd peels (Benincasa hispida). Rasayan Journal of Chemistry 2012, 5, 280-285.

How long did the vacuum evaporation of the extracts take and at what temperature?

For evaporating 95% ethanol, we aimed to use a lower temperature to minimize the risk of losing heat-sensitive compounds. Therefore, we used 40±5°C, which is below the boiling point of pure ethanol (approximately 78.37°C at standard atmospheric pressure) for 30 minutes. For 50% ethanol, we used 50±5°C for 60 minutes because the ethanol content is lower, making it less volatile.

Samples B95 and B50 were prepared by vacuum evaporation, DI by lyophilization. Explain why the results are considered comparable despite the different methods.

We focused on the influence of solvent polarity by using different organic solvents (95%v/v ethanol, 50%v/v ethanol, and water) in order to investigate the constituents in the peel that are responsible for the biological activities. To prevent the degradation of bioactive compounds, we chose the most suitable techniques for removing solvent. Lyophilization can be used for a wide range of sample types, including aqueous solutions so it is a suitable method for water removal. Both vacuum evaporation and lyophilization are effective strategies for minimizing chemical degradation in the specified solvents.

Why was ethanol chosen as a solvent next to water? On what basis did they decide the 95 and 50%? From an economic point of view, how do you judge the concentration of solvents? Is it possible that 75% would be enough?

   In this study, we were interested in a broad spectrum of compounds, so we had to use a mixture of water and ethanol to capture a wide range of polarities such as, quercetin, rutin, astilbin, catechin, naringenin, and hispidulin. Some compounds are more soluble in water, while others are more soluble in ethanol or other organic solvents. Therefore, we tried to adjust the ratio to maximize the solubility of the target compounds by increasing the ethanol content, which can help extract nonpolar compounds while increasing the water content can target more polar compounds. A lower ethanol concentration (50%) may extract a broader range of compounds from the plant material, including both polar and less polar compounds, so 50% ethanol may be a suitable choice. Furthermore, the cost of 75% ethanol may be higher than that of 50% ethanol. We considered safety and economic limits while reaching our decision.

Reference

Islam, M.T.; Quispe, C.; El-Kersh, D.M.; Shill, M.C.; Bhardwaj, K.; Bhardwaj, P.; Sharifi-Rad, J.; Martorell, M.; Hossain, R.; Al-Harrasi, A.; et al. A literature-based update on Benincasa hispida (Thunb.) Cogn.: Traditional uses, nutraceutical, and phytopharmacological profiles. Oxidative Medicine and Cellular Longevity 2021, 2021.

2.5.3. In what units were the FRAP results given?

FRAP value (mg FeSO4/g extract) = c*v*t/m

where C is the FeSo4 concentration (mg/mL) of the standard curve, V is the sample volume (mL), t is the dilution factor, and m is the weight of the sample dry (g).

Reference

Xiao, F.; Xu, T.; Lu, B.; Liu, R.H. Guidelines for antioxidant assays for food components. 2020.

2.5.4. What does the abbreviation APPH mean?

AAPH is a abbreviation of the chemical name which is 2,2'-azobis-(2-amidinopropane dihydrochloride, it has been added in Line 186. AAPH is a water-soluble azo compound that is used free radical generator, often in the study of lipid peroxidation and the characterization of antioxidants. The decomposition of AAPH produces molecular nitrogen and 2-carbon radicals. The carbon radicals may combine to produce stable products or react with molecular oxygen to give peroxyl oxidation.

References

Niki, E. Free radical initiators as source of water- or lipid-soluble peroxyl radicals. Methods Enzymol 1990, 186, 100-108.

Krishna, M.C.; Dewhirst, M.W.; Friedman, H.S.; Cook, J.A.; DeGraff, W.; Samuni, A.; Russo, A.; Mitchell, J.B. Hyperthermic sensitization by the radical initiator 2,2'-azobis (2-amidinopropane) dihydrochloride (AAPH). I. In vitro studies. Int J Hyperthermia 1994, 10, 271-281.

8 page, 353. és 354. lines: It was found that the B95 possessed the highest activity with the percentage inhibition of 53.42 ± 0.02, against peroxide radicals followed by the B50 and BW, respectively. BW and B50 are the correct order B95 extract was the best for inhibiting collagenase and hyaluronidase, even though it ranked first in various antioxidant assays only for ABTS and lipid peroxidation assay. In terms of DPPH, it ranks second and last in terms of FRAP. What mechanisms could be behind the fact that, despite the results, the antioxidant and anti-aging activities are still promising?

Antioxidant activities of the plant extracts can be described by various mechanisms including free radical scavenging, metal reducing property, inhibit chain breaking (inhibit peroxyl radicals). Each extract shows antioxidant properties through one or more than one mechanism due to chemical compounds in each extract. From this research, we focus on finding antioxidant and anti-aging properties of the extracts that related to skin aging. In addition, we focus on flavonoids, key compounds in extracts, that related with biological activities of the extract. Therefore, the B95 extract was chosen for further study because it has highest flavonoid contents with good antioxidant and anti-aging activities.

Reviewer 3 Report

The comments are listed below:

1. Line 68, Benincasa hispida fruit (A) and its peel that was used in investigation (B). → B. hispida fruit (A) and its peel (B) that was used in investigation.

2. Line 547, Benincasa hispida B. hispida

3. The fingerprint of B. hispida should be established and recognized the flavonoids, such as quercetin, rutin, astilbin, catechin, naringenin, and hispidulin in fingerprint.

4. In Figure 5, the concentration of B95, B50, and BW should be described.

Author Response

Thank you very much for taking the time to review our manuscript. Please find the detailed responses in the attached file and the corresponding revisions/corrections highlighted in the re-submitted files.

Comments

Responses

Line 68, Benincasa hispida fruit (A) and its peel that was used in investigation (B). → B. hispida fruit (A) and its peel (B) that was used in investigation

It has been already adjust following the comment from reviewer in Line 77.

Line 547, Benincasa hispida→ B. hispida

It has been already adjust following the comment from reviewer in Line 607.

The fingerprint of B. hispida should be established and recognized the flavonoids, such as quercetin, rutin, astilbin, catechin, naringenin, and hispidulin in fingerprint.

The method and results of HPLC analysis were obtained in 2.7 and 3.5 , respectively.

The most potential of B. hispida (B95) was selected to analysis and rutin was used as reference active compound to characterize B95.

In Figure 5, the concentration of B95, B50, and BW should be described.

It has been already adjust following the comment from reviewer in Figure 6.

Reviewer 4 Report

This paper studies the influence of Benincasa hispida peel extracts on antioxidant and antiaging activities and molecular docking simulation. The methods used in this study are specific, and the experimental procedure is very well explained. The experiments have been very well designed and carried out, and the conclusions drawn based on the experimental data are well-justified and highly valuable.

The authors can improve the manuscript.

In the introduction, it should be mentioned the data concerning the phenolic composition of Benincasa hispida extracts and their antioxidant activities that have already been published in the literature. The main objective of this study should be clearly stated, emphasizing the innovation of this work.

Author Response

Thank you very much for taking the time to review our manuscript. Please find the detailed responses in the attached file and the corresponding revisions/corrections highlighted in the re-submitted files.

Comments

Responses

In the introduction, it should be mentioned the data concerning the phenolic composition of Benincasa hispida extracts and their antioxidant activities that have already been published in the literature.

The information about potential phenolic compounds for antioxidant was added in Line 57-59.

The main objective of this study should be clearly stated, emphasizing the innovation of this work.

The main objective has already added in Line 68-75